# Resilience to periodic disturbances and the long-term genetic stability in *Acropora* coral
L. Thomas [1,2] ✉, D. Şahin[1,2], A. S. Adam[1], C. M. Grimaldi [1,2], N. M. Ryan [1], S. L. Duffy[1,2],
J. N. Underwood[1], W. J. Kennington [2,3] & J. P. Gilmour [1,2]

Climate change is restructuring natural ecosystems. The direct impacts of these events on biodiversity and community structure are widely documented, but the impacts on the genetic variation of populations remains largely unknown. We monitored populations of *Acropora* coral on a remote coral reef system in northwest Australia for two decades and through multiple cycles of impact and recovery. We combined these demographic data with a temporal genetic dataset of a common broadcast spawning corymbose *Acropora* to explore the spatial and temporal patterns of connectivity underlying recovery. Our data show that broad-scale dispersal and post-recruitment survival drive recovery from recurrent disturbances, including mass bleaching and mortality. Consequently, genetic diversity and associated patterns of connectivity are maintained through time in the broader metapopulation. The results highlight an inherent resilience in these globally threatened species of coral and showcase their ability to cope with multiple disturbances, given enough time to recover is permitted.

The capacity of natural populations to recover from acute disturbances and maintain genetic variation is a key aspect of their resilience[1–4]. The process of recovery is primarily influenced by rates of survival, growth and immigration[5,6]. When mortality is high and immigration is low, recovery depends on the few individuals surviving locally, and declines in genetic diversity are expected[7]. In contrast, when immigration is high and impacts vary among sub-populations, areas least affected can serve as a source of recruits to aid in the recovery more widely. In these cases, genetic diversity remains stable[8]. On tropical coral reefs, acute heat stress events and associated bleaching have become routine disturbances and are now the primary cause for coral mortality globally[9,10]. Monitoring patterns of recovery and associated changes in coral cover have informed how climate change is impacting coral reefs[11–13]; however, information on how fluctuating coral populations influence genetic variation are also required.

The isolated oceanic atolls of northwest Australia provide a unique opportunity to study coral populations through cycles of impact and recovery. The Scott system of three reefs is hundreds of kilometres from the mainland and neighbouring reefs, and is demographically isolated in space and time[14–18]. In recent decades, recurrent disturbances to the Scott reefs have caused dramatic shifts in coral population sizes and distribution[11,19–21].

From 1994 to 2021, moderate heat stress and tropical cyclones were frequent disturbances, reducing coral cover at one or more of our long-term monitoring (LTM) sites during 9 of the 27 years (Fig. 1). The most severe disturbances were mass bleaching events in 1998, and again in 2016, causing relative reductions in mean coral cover of ~75% following both events[11]. Cyclone Lua in 2012 also decreased mean cover by 8% across the reef system[22], with other smaller disturbances (e.g. bleaching in 2010) causing less severe and more localised impacts. The reefs had largely recovered 12 years after the 1998 mass bleaching, but with local variation in recovery depending on subsequent exposure to disturbances and the resilience of different coral groups within communities, leading to long-term shifts in community composition[11].

## Results and discussion

Through the regime of recent disturbances to the Scott system of reefs, the impacts of even the most severe event varied spatially, usually with a different group of sites worst affected by bleaching or by cyclones (Fig. 1b). Consequently, when *Acropora* cover decreased to very low levels (<1%) at some sites following disturbance, it was often higher (1–7%) at other sites across the reef system (Fig. 1b). This spatial variation in cover of *Acropora*

¹Australian Institute of Marine Science, Indian Ocean Marine Research Centre, Crawley, Australia. ²UWA Oceans Institute, The University of Western Australia, Crawley, Australia. ³Centre for Evolutionary Biology, School of Animal Biology, The University of Western Australia, Perth, Australia.
✉e-mail: l.thomas@aims.gov.au

reflected the changes in abundance of adult colonies (>20 cm) within the long-term monitoring sites (Fig. 1c).

The loss of adult colonies caused a comparative reduction in reproductive output and larval supply to recruitment tiles across the reef system ($R^2$ = 0.857; red points, Fig. 2a); however, this stock-recruitment relationship was much weaker at the scale of the individual sites (200 m$^2$), with the abundance of adult colonies being a poor predictor of larval supply to settlement tiles locally ($R^2$ = 0.438; purple points, Fig. 2a). Subsequent recovery at each site was then driven by local conditions, with the supply of *Acropora* larvae to settlement tiles predicting ($R^2$ = 0.706) the number of new *Acropora* recruits (<2 cm) on natural substrata six to 12 months later (Fig. 2b). Survival of these recruits and the resulting juveniles (6–15 cm) was also high (Fig. 2c), with the number of recruits on substrata influencing the number of juvenile colonies within each site 12–24 months later (Fig. 2d).

Demographic changes in *Acropora* assemblages provided indirect evidence of recovery at the Scott reefs being seeded by the supply of larvae from less affected sites. This mechanism of recovery was supported by population genetics of a common species of corymbose *Acropora*.

Low-coverage whole genome re-sequencing of *Acropora sp.* (formerly *Acropora tenuis*[23]) revealed high gene flow and connectivity across the Scott system of reefs, with no clustering of samples by site using a genotype likelihood approach based on 8,329,442 variant sites (Fig. 3a). This pattern of high gene flow was confirmed using more stringent depth filters (29,462 single nucleotide polymorphisms (SNPs); Fig. S1). Mean $F_{ST}$ was −0.0002 (Fig. S1) and most pairwise estimates between sites were negative and few were significant (Fig. 3b). The exception was the sample site from Seringapatam reef (SS3), which showed low ($F_{ST}$ < 0.004) but significant levels of genetic differentiation with all other sites from North and South Scott (Fig. 3b, Supplementary Data 1). Together, the whole genome re-sequencing data pointed to broadscale dispersal and high gene flow across the reef system.

Our temporal genetic dataset of microsatellite genotypes (2004, 2009, 2014, 2015, 2021) revealed a consistent pattern of connectivity underlying recovery (Fig. 3c). Within each year, mean estimates of genetic differentiation were negative, and most pairwise values were low and non-significant (Fig. S2, Supplementary Data 2). The genetic patchiness that did arise

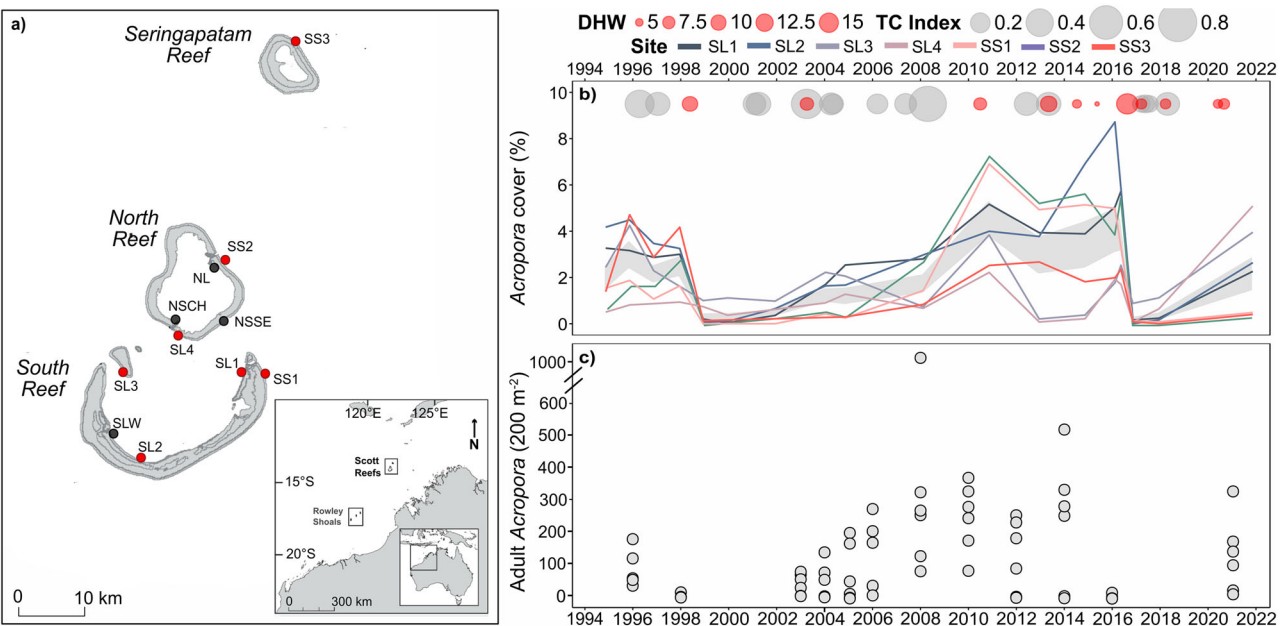

**Fig. 1 | Periodic disturbances and mortality on the Scott system of reefs.**
**a** Permanent monitoring sites in which demographic (red symbols) and genetic variation (red and black symbols) were assessed through time. **b** Changes in the
percentage cover (± se mean grey bar) of corymbose *Acropora;* and **c** density of adult *Acropora* colonies (with each grey circle representing an LTM site).

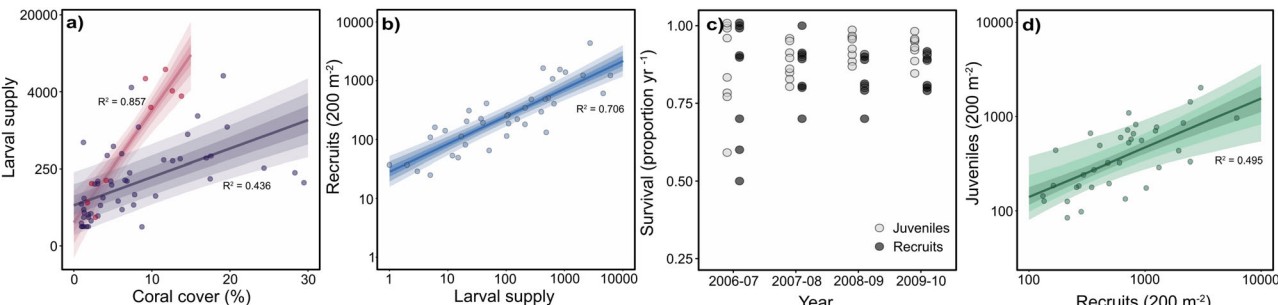

**Fig. 2 | Broadscale dispersal and post-recruitment survival drives recovery.**
**a** Stock-recruitment relationship for *Acropora* at each site (purple, 200 m$^2$) and across the reef system (red, years). Larval supply is the number of settlers on tiles following mass-spawning. **b** Relationship between larval supply to recruitment tiles and density of new *Acropora* recruits (≤2 cm) on substrata, 6 to 12 months later, within each site (200 m$^2$). **c** Annual survival of *Acropora* recruits (<5 cm) and
juveniles (6–15 cm) at sites during years of rapid recovery (2006–2010) following mass bleaching (left). **d** Relationship between density of *Acropora* recruits and juveniles, 12 to 24 months later, within each site (200 m$^2$). Dark lines are the median slope and coloured ribbons (darker to lighter) show 50%, 75%, and 95% posterior density intervals. Conditional $R^2$ values are displayed for each model.

within each timepoint was ephemeral and changed from year to year (Fig. S3). As a result, there were no significant differences in allele frequencies among sites nested within years ($\theta_{ST} = -0.008$, $P = 0.752$). Differences among years under the AMOVA framework accounted for only 1.6% of the total variation ($\theta_{RT} = 0.0192$, $P = 0.003$) and we did not observe any clustering of samples by year in multi-dimensional space (Supplementary Data 3, Fig. 3c). This pattern of chaotic genetic patchiness is a common attribute of broadcast spawning marine species, where the complex interplay between biological and physical processes often drives unstable patterns of spatial genetic structure[24,25]. In turn, one generation of patches does not predict the next (Fig. S4). These levels of genetic differentiation among sites and years were outside the bounds of normality when we randomized site-time assignments and recalculated genetic differentiation (Fig. S5), suggesting that they more likely reflect stochastic biological processes than technical artefacts.

Despite recurrent disturbances and dramatic demographic changes in *Acropora* assemblages, the genetic diversity of the broader metapopulation at the Scott reefs has remained stable. We did not observe any changes in allelic richness or heterozygosity among our sampling timepoints (Fig. 4a, b; Supplementary Data 4). While site-level estimates seemed to vary slightly among years, genetic diversity in the broader population remained stable across our study period. Although we did not have samples from the Scott reefs that predated the 1999 mass bleaching event, we found that levels of genetic diversity were similar to samples collected from the neighbouring Rowley Shoals, an offshore oceanic atoll system 400 km to the south of the Scott reefs that has avoided widespread bleaching and mortality over similar timescales[11]. Indeed, microsatellite markers (Fig. 4a, b, Supplementary Data 4) and whole genome re-sequencing techniques (Fig. 4c, d) indicated that genetic diversity in the two systems are similar, despite strong genetic drift (genome-wide $F_{ST} = 0.078$, Fig. S6) and contrasting disturbance

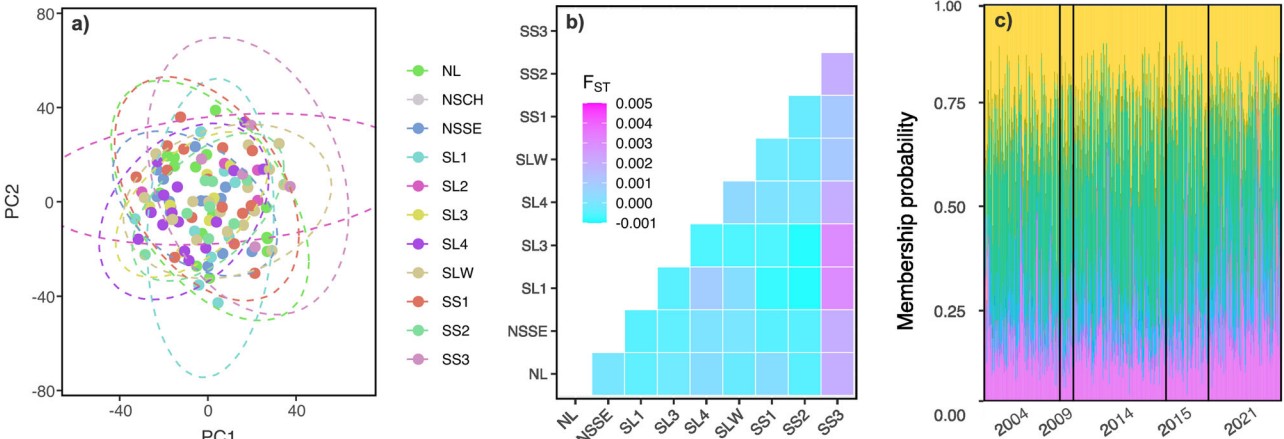

**Fig. 3 | High genetic connectivity across the Scott system of reefs. a** Scatter plot of the first two principal components for samples collected in 2021 and based on 8,329,442 variant sites using a whole genome genotype likelihood approach. Each point represents a unique coral colony and is colour coded by site. Dashed lines represent 95% confidence ellipses for each site. **b** Tiled heatmap of genetic differentiation (pairwise $F_{ST}$ Weir and Cockerham, 1984) among sites in 2021 using a reduced dataset of 29,462 single nucleotide polymorphisms with more strict depth filters (depth > 10). Warmer colours indicate higher values (Supplementary Data 1). **c** Genotype composition plot of assignment probabilities for individuals (vertical bars) to predefined groups based on year of collection (K = 5) using the temporal genetic dataset of microsatellite genotypes.

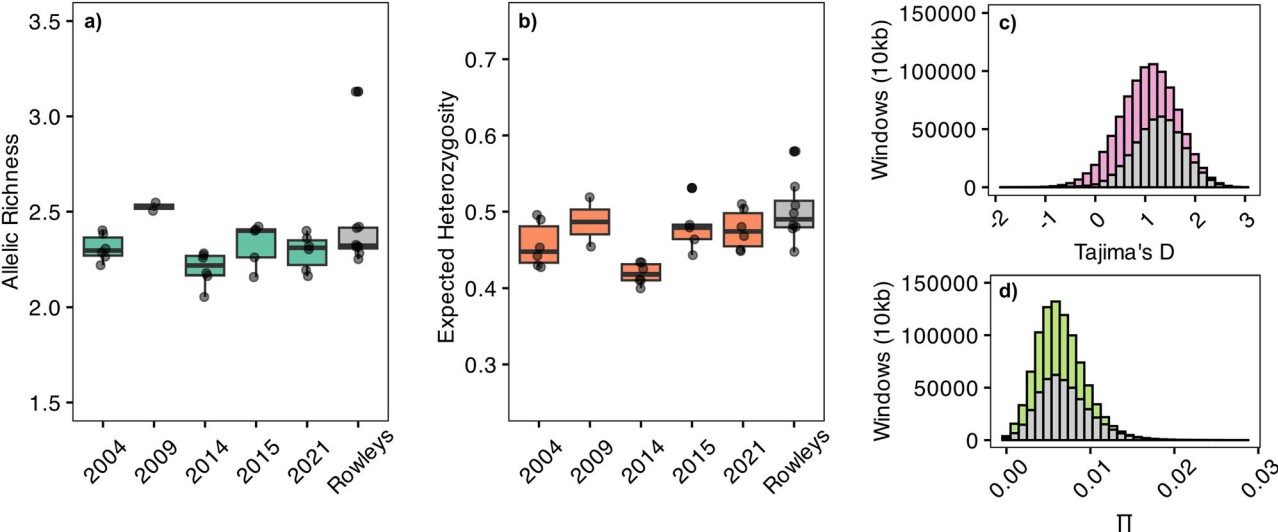

**Fig. 4 | Genetic diversity is stable through time. a** Boxplots of allelic richness and **b** expected heterozygosity at each timepoint based on the microsatellite genotype data. Each black point represents a site value averaged across five loci. The boxes represent the interquartile range (IQR) of the data distribution, and the thick black line inside the box represents the median. Data from the Rowley Shoals are also provided for regional context. **c** Histograms of Tajima's D and **d** nucleotide diversity (Π) for Scott Reef (coloured) and Rowley Shoals (grey) based on whole genome resequencing data using a genotype likelihood approach.

histories. Thus, despite severe declines in coral cover, our data suggest that there have been enough survivors at Scott system of reefs to facilitate recovery and maintain moderate levels of genetic diversity in the metapopulation.

The persistence of coral populations depends largely on their connectivity and larval supply, and their capacity to absorb and adapt to emerging pressures. Our results suggest that a high dispersal capacity and post-recruitment survival in *Acropora* are key mechanisms of resilience to climate change. This is in contrast to species with a lower capacity to disperse across a group of reefs, such as some brooding corals or those reliant on vegetative propagation[26]. However, our data also highlight that recovery depends on spatial variation in impacts and relatively high abundances of survivors at several locations across the reef system, in addition to considerable time between severe disturbances. Under these conditions, population abundances and genetic diversity of *Acropora* corals may be maintained, but with an increasing severity and frequency of disturbances through global heating, longer term degradation is likely, particularly at these isolated coral reefs. This study underscores the importance of long-term monitoring to understand the impacts of climate change on coral reef ecosystem health. More importantly, it highlights an inherent resilience in broadcast spawning coral populations on remote reefs, and offers a glimmer of hope to coral reef ecosystems more broadly against a backdrop of rapid environmental change.

## Methods

### Demographic changes in *Acropora*

In 1994, permanent transects (250 metres) were established at 21 reef slope sites replicated across seven locations and three reefs[27]. At each location (Fig. 1a), three sites were separated by approximately 300 metres and consisted of 250 metres of permanent transects marked at 10-m intervals. Surveys were conducted annually between 1994 and 1999, and then in 2003, 2004, 2005, 2008, 2010, 2012, 2014, 2016, 2017, and 2021. In 2016, additional surveys were conducted in January, April and October 2016, before, during and after the mass bleaching, and from 2016 only the first site at each location was surveyed to allow for expansion into additional habitats (not included here). During each survey, a tape was laid along the permanent transect and images of the benthic community captured from a distance between 30 and 50 cm from the substrata. Images were analysed using point sampling technique and benthic groups identified to the lowest taxonomic resolution achievable by each observer[28]. These data were then divided among benthic groups according to taxa (e.g., family, genus) and growth form (e.g. encrusting, foliose, massive, branching).

### Larval supply and recruitment

Larval supply of the most diverse and abundant coral genus, the *Acropora*, was quantified using terracotta settlement tiles along the permanent transects at 18 sites at six of the reef slope locations, during nine years between 1996 to 2013[21] (1996–1999, 2002, 2003, 2008, 2010, 2011). At each of the sites, groups of six settlement tiles (10 cm × 10 cm) were deployed at distances of 30–100 cm apart, spaced at 60 m along the permanent transect. Settlement tiles (*n* = 108 tiles total) were deployed and collected one month either side of the primary mass coral spawning[29]. Size-frequency distributions of *Acropora* colonies (excluding staghorn and hispidose growth forms) were quantified along the permanent transects (200 m) at the first site at each of six reef slope locations. Colony sizes (longest linear dimension) were recorded to the nearest centimetre along ten 20 m transects spaced at 10 m intervals, within a width of 25 cm for colonies <10 cm (50 m²) and 1 m for colonies ≥10 cm (200 m²). Density at each site was adjusted to 200 m². Growth and survival of colonies was quantified in a species of corymbose *Acropora* (nominally *Acropora spicifera*) at two of the permanent sites at four reef slope locations over four years (2006–2010). Locations were chosen to be representative of the primary habitat for corymbose *Acropora* across the reef system. Colonies (*n* = 3692; of all sizes were tagged and resurveyed annually. Given issues identifying juvenile coral to species, an assemblage of

*Acropora* species were tagged for colonies <10 cm in size. Size and growth were quantified from photographs taken directly above the colony, or perpendicular to the maximum length when first tagged. Colonies were digitized and their size defined by the maximum diameter through the ellipse of best fit.

### Microsatellite genotyping

Tissue samples were collected from 982 colonies of a corymbose *Acropora* (*Acropora sp.*) at five timepoints (2004, 2009, 2014, 2015, and 2021; Supplementary Data 5) spanning 17 years. At each site, samples were taken from colonies separated by at least 1.5 m and not from loose fragments or colonies directly down-slope from another colony. Genomic DNA was extracted using a Qiagen DNeasy kit or using a high-throughput membrane-based DNA extraction protocol[30] and genotyped across at seven microsatellite loci[16,31] in 10 μL multiplex PCR reactions using the Qiagen multiplex PCR kit. Fragment analysis was carried out on a GE Healthcare MegaBACE 1000 capillary sequencer or an Applied Biosystems 3730 capillary sequencer. Electropherograms were visualised and scored using the software GeneMarker v1.91 (SoftGenetics). To facilitate comparison of allele sizes among collections that were analysed on different platforms, 19 control samples from earlier collections that exhibited a representative range of allele sizes were run alongside the new samples and allele sizes calibrated accordingly. To further minimise genotyping errors, automated scorings of alleles in all microsatellite analyses were checked manually, and uncertainties were cleared by re-amplification and comparison. Of the seven loci originally trialled, two loci scored in 2004 and 2009 (Amil2_006 and Amil5_028) could not be consistently scored and/or calibrated on the Applied Biosystems sequencer, and these loci were removed.

### Cryptic lineages and clonality

Previous data from the Scott reefs have shown cryptic lineages are widespread in *Acropora* and reflect divergence in reproductive timing, with the main spawning period during austral autumn and a secondary smaller event in austral spring[29,32]. Sympatric colonies of *Acropora sp.* display strong genetic divergence, suggesting pronounced reproductive boundaries and limited gene flow[29,33–35]. Before analysing patterns of genetic variation through space and time, an initial screening of all samples using microsatellite data was conducted to remove the less abundant spring spawning lineage to focus our analyses on a single gene pool. To do this, we utilized data from 2009 that tracked a subset of colonies (*n* = 68) across multiple years with a consistent seasonal spawning phenotype[29]. These samples were used to assign samples from all timepoints to one of the two lineages using model-based Bayesian clustering in Structure[36] (Fig. S7). We ran *structure* with the *no prior* model at K = 2 based on the panel of five common microsatellites. Analyses were run ten times using the admixture model with independent allele frequencies and with a burn-in period of 100,000, followed by 500,000 MCMC replications for each run. These model conditions were implemented because the genetic differentiation between spawning groups is large and initial investigations showed that runs with correlated allele frequencies model overestimated K and individuals lacked strong assignment[29]. After removing the spring spawning lineage and poor-quality samples that failed to amplify, we were left with 506 autumn spawning colonies (321 multi-locus genotypes) with greater than 0.75 probability of assignment (mean assignment was 0.96 +/− 0.05 S.D.). Genotypic richness in the targeted species at the Scott reefs is high (>0.96[15]) with little evidence of clonal propagation via vegetative fragmentation; however, replicate multi-locus genotypes were common in our temporal dataset when we restricted analyses to the five microsatellite loci that could be scored across all years. The 2004 and 2009 datasets included two additional loci[15] and showed that the replicate MLGs in the temporal dataset were indeed unique individuals, and so we proceeded with a dataset of all 506 multi-locus genotypes for downstream analyses. We also clone corrected at the population level using *poppr* and confirmed patterns identified with the larger dataset held true when focussing on unique MLGs.

## Genetic differentiation and diversity

Using our temporal dataset of microsatellite genotypes for the autumn spawning lineage, we calculated pairwise estimates of $F_{ST}$ among sites with *stammp*[37] and adjusted for multiple comparisons using a False Discovery Rate in R (*p.adjust*). We used *prcomp* (R Core Team, 2021) to carry out principal components analyses and used Discriminant Analyses of Principal Components (DAPC) with *adegenet*[38] to estimate group assignment probabilities under a Bayesian framework. To explore hierarchical patterns of genetic structure, we carried out an analysis of molecular variance (AMOVA) with sample sites nested within timepoints. To measure levels of genetic diversity, we calculated allelic richness and expected heterozygosity for each site and timepoint comparison using *hierfstat*[39]. We then used a Bayesian generalised linear mixed effects model with a gaussian distribution to test for differences in genetic diversity among years. Sampling site was included as a random effect to account for the lack of spatial independence. The model was based on three chains with 3000 iterations, including 1000 iterations to warm-up and a thinning interval of 5. All Bayesian models were performed in Stan (Stan Development Team, 2021) using the "brms" package[40]. We ran posterior predictive checks and visually inspected all models for violations for statistical assumptions using the 'dharma' package (Hartig, 2022). Finally, to compare these estimates of genetic diversity with a pristine reef system, we genotyped conspecific colonies of *Acorpora sp.* from the Rowley Shoals using the same panel of microsatellite markers ($n = 196$ colonies) and whole genome re-sequencing techniques ($n = 10$ colonies, below).

## Whole genome re-sequencing

We re-sequenced the genomes of samples ($n = 111$ colonies) collected in 2021 to approximately 8X coverage (Fig. S8) using Covaris library preparation via sonication and sequenced on a NovaSeq at Genomics WA. Raw sequence files were mapped with *bwa mem* (Supplementary Note 1) to the *A. tenuis* pseudo-chromosome-scale assembly (*aten.chr.fasta*)[33]. We used *samtools*[41] to sort and index and *picardtools* (http://broadinstitute.github.io/picard) to mark and remove duplicate reads. *Samtools*[41] was used to calculate sequencing depth of each sample at each position across the genome (Supplementary Note 2, Fig. S10). We called single nucleotide polymorphisms (SNPs) using *mpileup* in *bcftools*[42] and filtered using *samtools* and based on a minor allele frequency of 0.05, a call rate of 0.95, and a mean read depth of 10x per sample (Supplementary Note 3). This produced a variant call file of 29,462 biallelic SNPs that was used to estimate pairwise $F_{ST}$ among sites (minimum of 5 individuals per site) using *stamp*. We also explored genetic structure among sites using a genotype likelihood approach in *angsd*[43] and estimated genome-wide allele frequencies to carry out principal components analyses using *pcangsd*[44] (Supplementary Note 4). We masked sites with a minQ < 20, minMapQ < 30, sites with less than 1/3 and greater than 2x the mean read depth and removed loci with missing data in more than half of the samples. We only retained variable sites (snp_pval < 1e−6) and those with a minor allele frequency greater than 0.05. This filtering criteria produced 8,329,442 variant sites used for clustering analysis. We also re-sequenced a subset of genomes from the Rowley Shoals ($n = 10$) using Nextera Flex Library Preparation Kits (Illumina) and sequenced on a NovaSeq.

## Reporting summary

Further information on research design is available in the Nature Portfolio Reporting Summary linked to this article.

## Data availability

All of the data to support the findings of this study can be found on the Open Science Framework https://doi.org/10.17605/OSF.IO/2VBHW (https://osf.io/2vbhw/)[45].

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

## Acknowledgements
This research was funded by the Australian Institute of Marine Science and Woodside Energy Ltd as Operator for and on behalf of the Browse Joint Venture (BJV). We would like to thank the crew of the RV Solander for their professionalism and hard work. We gratefully acknowledge the support from the Australian Cancer Research Foundation for the Centre for Advanced Cancer Genomics which has made available the Illumina Novaseq6000 systems for the use of Genomics WA. We also thank Helix Solutions for their support with microsatellite genotyping. This manuscript was written on Whadjuk Noongar *boodja*. We acknowledge the Whadjuk people as traditional owners of this land and pay respect to their elders past and present.

## Author contributions
J.G., L.T., and J.U. designed the study, collected samples, analysed the data. D.S., S.D., N.R. and C.G collected samples and analysed data. A.S. and W.J.K. analysed data. All authors wrote the manuscript.

## Competing interests
The authors declare no competing interests.
