## [Peer Review File · Communications Biology]

Reviewers' comments:

Reviewer #1 (Remarks to the Author):

This paper demonstrates that coral sites are genetically connected and that widescale dispersal of larvae is critical to regional reef recovery. Overall, this is a very interesting paper with valuable information to share with the broader community. I have three bigger picture comments. They do not take away from the paper, but rather are intended to provide additional depth and transparency to the interpretation. First, the results are based only on *Acropora*. So, the interpretation should be limited to *Acropora* with a suggestion that it could apply more broadly with additional research. Second, it seems worth exploring more directly that it appears that recovery is a function of the % cover of surviving adults. While this seems obvious, it is glossed over in the text. Finally, the paper concludes that recovery of reefs depends on at least a decade between severe disturbances. However, the data set is only 21 years long with only one documented decadal-scale recovery event. I realize that this is a very long, and comprehensive data set with a lot to offer. But technically, it is not possible to know if you need at least a decade between large events when you only have one decadal-scale recovery event documented. I think this conclusion should be couched more as a suggestion. Additional specific comments are included as comments in the uploaded manuscript text. Additional minor comments are below.

- Fig S1: brackets appear to be missing.
- Fig S2, S9, and 3: Would be good to indicate which sites belong to each reef in each figure legend.
- Fig S3: vertical line not visible
- Fig S5: seems there is supposed to be two panels for this figure, but only one is presented.
- Tables S1-S5: uncertain which table heading goes with which excel file as the excel files are not labeled with table number.
- Fig 1: define LTM, DHW, TC Index; in b) lines and shading appear green (not gray). Hard to see changes in *Acropora* that correspond to TC (I assume these are the damaging waves?). It is not possible to follow the trajectory of any one site in (b) and (c) as all sites are the same color. See additional comments in text. Is there a statistical test to support the statement "changes in percentage cover...and density... were driven by exposure to heat stress...or damaging waves"? If so, include a reference to that statistical result in the figure legend. If not, remove this conclusion sentence from the figure legend.
- Fig 2: if there are 90 tiles, why are there fewer than 90 points on each graph?
- Fig 4: title states that genetic diversity is stable through time, but allelic richness and expected heterozygosity in 2009 looks very different than in 2014.

Reviewer #2 (Remarks to the Author):

The manuscript addresses the question, can a coral population experiencing severe, periodic disturbances maintain its genetic diversity and hence resiliency to climate change over time? This topic should be of interest to a wide community beyond just coral biologists. The study is based on an unusually comprehensive data set spanning the period 1994-2016 for a remote reef system off the NW coast of Australia consisting of an extensive set of demographic data and temporal genetic data. Analysis of the data allows the authors to make the exciting discovery that coral populations can maintain genetic diversity and community stability over two decades despite numerous severe bleaching/mortality events. The key finding was that recovery depends on spatial variation in impacts and relatively high abundances of survivors at several locations across the reef system, in addition to at least a decade between severe disturbances.

I find the findings convincing and the methodology rigorous. I do not feel qualified to comment on the appropriateness of the methods used to compare allele richness between early and later parts of the

study that relied on use of different platforms (lines 157-167). Obviously this is important. The way they describe sounded reasonable to me.

Reviewer #3 (Remarks to the Author):

I really like this paper, the authors have done a great job drawing meaningful and relevant conclusions from the data that at the first glance don't show highly exciting patterns. It is a really well done study and one of the very few (if not the only one thus far, my knowledge here might be incomplete) reports of temporal change in coral population genetics, albeit only with microsatellites.

I have just one concern/comment, about chaotic genetic patchiness. This is a very interesting observation! I'm only aware of a single coral paper claiming the evidence for "sweepstakes reproductive success" in corals (Barfield et al 2022). This is pretty fundamental for coral biology so I'd recommend emphasizing this more. That said, can you be sure that the pairwise F_{st} variation you are seeing (Fig S5) is not simply due to sampling noise? One way to check this is to randomize population:time assignments and redoing the F_{st} heatmap.

Comments to Reviewers

Resilience to periodic disturbances and the long-term genetic stability in Acropora coral

Below you will find an itemised list of the reviewer comments (black font) along with the corresponding author response (red font). Thank you for your time in reviewing this manuscript, it is much appreciated. We hope that you are satisfied with our revised submission.

Dear Dr Thomas,

Your manuscript entitled "Resilience to periodic disturbances and the long-term genetic stability in coral" has now been seen by 3 referees. You will see from their comments below that while they find your work of considerable interest, some important points are raised. We are interested in the possibility of publishing your study in Communications Biology, but would like to consider your response to these concerns in the form of a revised manuscript before we make a final decision on publication. We therefore invite you to revise and resubmit your manuscript, taking into account the points raised. In particular, reviewer #1 suggests adding caveats to the conclusions given the time constraints of the study. Additionally, please respond to reviewer #3 regarding the robustness of the Fst analysis.

Please highlight all changes in the manuscript text file.

Reviewers' comments:

Reviewer #1 (Remarks to the Author):

This paper demonstrates that coral sites are genetically connected and that widescale dispersal of larvae is critical to regional reef recovery. Overall, this is a very interesting paper with valuable information to share with the broader community. I have three bigger picture comments. They do not take away from the paper, but rather are intended to provide additional depth and transparency to the interpretation.

Thank you for the nice comments on our paper, we are pleased that you found it interesting and a valuable contribution to the field.

First, the results are based only on Acropora. So, the interpretation should be limited to Acropora with a suggestion that it could apply more broadly with additional research.

We appreciate this concern and have now revised the language that we use throughout the manuscript to clarify that our results are limited to this group of coral. We have also included Acropora in the title to make this abundantly clear from the get-go.

Second, it seems worth exploring more directly that it appears that recovery is a function of the % cover of surviving adults. While this seems obvious, it is glossed over in the text.

Our data shows that the stock recruitment relationship was strong at the reef-level and broke down at the site-level (Lines 62-65). While the relationship at the site level was still significant, the relationship was very weak. We have included a clarification sentence in the results section highlighting that recovery seems to be a function of % cover of surviving adults at the reef scale, but that this stock recruitment relationship breaks down at smaller spatial scales (Lines 61-70).

*'This loss of adult colonies caused a comparative reduction in reproductive output and larval supply to recruitment tiles across the reef system ($R^2 = 0.857$; red points, Fig. 2a). Together, these data indicate that recovery across the reef was a function of the percent cover of surviving adult colonies, however, this stock-recruitment relationship was much weaker at the scale of the individual sites (200m^2), with the abundance of adult colonies being a poor predictor of larval supply to settlement tiles locally ($R^2 = 0.438$; purple points, Fig. 2a). Subsequent recovery at each site was then driven by local conditions, with the supply of *Acropora* larvae to settlement tiles predicting ($R^2 = 0.706$) the number of new *Acropora* recruits (<2cm) on natural substrata six to 12 months later (Fig. 2b). Survival of these recruits and the resulting juveniles (6-15 cm) was also high (Fig. 2c), with the number of recruits on substrata influencing the number of juvenile colonies within each site 12-24 months later (Fig. 2d).'*

Finally, the paper concludes that recovery of reefs depends on at least a decade between severe disturbances. However, the data set is only 21 years long with only one documented decadal-scale recovery event. I realize that this is a very long, and comprehensive data set with a lot to offer. But technically, it is not possible to know if you need at least a decade between large events when you only have one decadal-scale recovery event documented. I think this conclusion should be couched more as a suggestion.

Thank you for this comment. We agree that perhaps some of our conclusions were overstated around windows for recovery. We have now removed this conclusion and toned down the language around time it takes for recovery throughout the manuscript.

Additional specific comments are included as comments in the uploaded manuscript text. Additional minor comments are below.

- Fig S1: brackets appear to be missing. This has been corrected and removed from figure legend

- Fig S2, S9, and 3: Would be good to indicate which sites belong to each reef in each figure legend. This has been corrected and added to the figure caption

- Fig S3: vertical line not visible

This has been corrected and added to the figure (now Fig S1).

- Fig S5: seems there is supposed to be two panels for this figure, but only one is presented.

This has been corrected and Figure legend updated.

- Tables S1-S5: uncertain which table heading goes with which excel file as the excel files are not labelled with table number. Table headers are now provided in Row 1 of each Table file to avoid confusion.

- Fig 1: define LTM, DHW, TC Index; in b) lines and shading appear green (not gray). Hard to see changes in *Acropora* that correspond to TC (I assume these are the damaging waves?). It is not possible to follow the trajectory of any one site in (b) and (c) as all sites are the same color. See additional comments in text. Is there a statistical test to support the statement "changes in percentage cover...and density... were driven by exposure to heat stress...or damaging waves"? If so, include a reference to that statistical result in the figure legend. If not, remove this conclusion sentence from the figure legend. Figure 1 has been updated to include site-level information for panel 1b. In addition, the conclusion in the figure legend was not supported by any statistical test and so we have removed this from the legend.

- Fig 2: if there are 90 tiles, why are there fewer than 90 points on each graph?

This was a typo. Each purple point in Figure 2a represents the total number of recruits found on tiles at each site (16 tiles per site, 108 per year). We have now clarified this in the methods (Lines 299-304) and the Figure 2 legend.

‘Larval supply of the most diverse and abundant coral genus, the *Acropora*, was quantified using terracotta settlement tiles along the permanent transects at 18 sites at six of the reef slope locations, during nine years from 1996 to 2013²¹ (1996-1999, 2002, 2003, 2008, 2010, 2011). At each of the sites, groups of six settlement tiles (10cm x 10cm) were deployed at distances of 30-100cm apart, spaced at 60 m along the permanent transect. Settlement tiles (n = 108) were deployed and collected one month either side of the primary mass coral spawning at the Scott reefs³⁴.’

- Fig 4: title states that genetic diversity is stable through time, but allelic richness and expected heterozygosity in 2009 looks very different than in 2014.

Measurements of genetic diversity in 2009 (allelic richness, He, Fis) were similar to other sites and timepoints and not statistically different.

Reviewer #2 (Remarks to the Author):

The manuscript addresses the question, can a coral population experiencing severe, periodic disturbances maintain its genetic diversity and hence resiliency to climate change over time? This topic should be of interest to a wide community beyond just coral biologists. The study is based on an unusually comprehensive data set spanning the period 1994-2016 for a remote reef system off the NW coast of Australia consisting of an extensive set of demographic data and temporal genetic data. Analysis of the data allows the authors to make the exciting discovery that coral populations can maintain genetic diversity and community stability over two decades despite numerous severe bleaching/mortality events. The key finding was that recovery depends on spatial variation in impacts and relatively high abundances of survivors at several locations across the reef system, in addition to at least a decade between severe disturbances. I find the findings convincing and the methodology rigorous. I do not feel qualified to comment on the appropriateness of the methods used to compare allele richness between early and later parts of the study that relied on use of different platforms (lines 157-167). Obviously this is important. The way they describe sounded reasonable to me.

Thank you for the positive comments.

Reviewer #3 (Remarks to the Author):

I really like this paper, the authors have done a great job drawing meaningful and relevant conclusions from the data that at the first glance don't show highly exciting patterns. It is a really well done study and one of the very few (if not the only one thus far, my knowledge here might be incomplete) reports of temporal change in coral population genetics, albeit only with microsatellites.

Thank you for the kind words.

I have just one concern/comment, about chaotic genetic patchiness. This is a very interesting observation! I'm only aware of a single coral paper claiming the evidence for "sweepstakes reproductive success" in corals (Barfield et al 2022). This is pretty fundamental for coral biology so I'd recommend emphasizing this more. That said, can you be sure that the pairwise F_{st} variation you are seeing (Fig S5) is not simply due to sampling noise? One way to check this is to randomize

population:time assignments and redoing the Fst heatmap.

This is indeed an interesting point and a great suggestion. We have gone back to the data and carried out some analyses as the reviewer suggests, primarily to determine whether the patches that we do see pop up in the data reflect biological patterns of not technical artefacts (Lines 94-98, Fig. S5). ‘Further analyses revealed that the observed levels of genetic differentiation among sites and years were outside the bounds of normality when we randomized site-time assignments and recalculated genetic differentiation (Fig. S5), suggesting that they more likely reflect stochastic biological processes than technical artefacts.’

Figure S5 Results from randomization trials where site:year assignments were shuffled and genetic differentiation among sites and years recalculated. Left: histogram of number of significant pairwise FST comparisons among sites across 100 bootstrap replicates. Right: histogram of p-values for estimates of genetic variation among timepoints under the AMOVA framework after 100 bootstrap resampling replicates. Vertical black lines denote 95% tails of the randomized distribution and blue lines represent the observed values.

REVIEWERS' COMMENTS:

Reviewer #3 (Remarks to the Author):

I am happy to see that the only concern that I had has now been fully addressed. Great work.